# Obstetric and Neonatal Outcomes in Mild Idiopathic Polyhydramnios

**DOI:** 10.3390/children9111624

**Published:** 2022-10-26

**Authors:** Lucia Pasquini, Ilaria Ponziani, Marta Pallottini, Giulia Masini, Viola Seravalli, Carlo Dani, Mariarosaria Di Tommaso

**Affiliations:** 1Fetal Medicine Unit, Department for Women and Child Health, Careggi University Hospital, 50134 Florence, Italy; 2Department of Health Sciences, Division of Obstetrics and Gynecology, University of Florence, 50134 Florence, Italy; 3Division of Neonatology, Careggi University Hospital of Florence, 50134 Florence, Italy; 4Department of Neurosciences, Psychology, Drug Research and Child Health, Careggi University Hospital of Florence, 50134 Florence, Italy

**Keywords:** mild polyhydramnios, idiopathic, maternal outcomes, perinatal outcomes, Cesarean section

## Abstract

*Background:* Idiopathic polyhydramnios is a controversial clinical condition, as data on perinatal outcomes are conflicting and vary depending on the severity of the condition. The aim of the present study was to compare obstetric and neonatal outcomes between pregnant women with mild idiopathic polyhydramnios and a control population. *Methods*: A retrospective cohort study was performed at a single university hospital comparing the obstetrics and neonatal outcomes of pregnancies with mild idiopathic polyhydramnios (*n* = 109) and control pregnancies (*n* = 2550). *Results*: Cesarean section (CS) was significantly increased in the group with polyhydramnios compared to controls (46% vs. 32%, respectively, *p* = 0.047) due to a higher rate of emergency CS in the polyhydramnios group (*p* = 0.041) because of abnormal cardiotocography (7.3% vs. 2.9%; *p* = 0.018) or labor dystocia (8.2% vs. 2.9%; *p* = 0.006). No statistically significant difference was found in the Apgar score, in the rate of neonatal hypoxia, or in the incidence of macrosomia between groups. In four cases, additional diagnoses of anomalies were made after birth, with a rate of 3.2%, which is comparable to the general population. *Conclusion*: Besides an increased risk of CS, patients with mild idiopathic polyhydramnios should be reassured regarding maternal and feto-neonatal outcomes. The management of pregnancies with stable mild idiopathic polyhydramnios should not differ from uncomplicated pregnancies, except for the need for increased labor surveillance.

## 1. Introduction

Polyhydramnios is an abnormal increase in the amniotic fluid volume measured by ultrasound and complicates 1–2% of singleton pregnancies [1], although in some studies the reported prevalence is even higher (up to 3.9%) [2,3]. It is defined as the presence of a single deepest vertical pocket (DVP) of amniotic fluid ≥8 cm or an amniotic fluid index (AFI) ≥25 cm [4]. Based on the AFI, polyhydramnios is generally categorized into three subtypes: mild (25.0–29.9 cm), moderate (30.0–34.9 cm), and severe (≥35.0 cm) [5].

The excess amniotic fluid can be associated with different conditions, such as gestational and nongestational diabetes, congenital fetal malformations, chromosomal abnormalities or genetic syndromes, fetal anemia, placental abnormalities, and fetal infections. If a cause of amniotic fluid increase is not identified antenatally, polyhydramnios is defined as idiopathic and, according to the most recent studies, this type accounts for 60–70% of all polyhydramnios cases [1,6,7]. 

Before defining polyhydramnios as idiopathic, all other above-mentioned conditions should be carefully ruled out, especially gestational diabetes, which is the most frequent. As reported in a recent study, an adequate and universal screening for gestational diabetes should be recommended and followed by appropriate counseling regarding this condition in order to prevent possible complications such as polyhydramnios [8].

High rates of maternal and neonatal complications are reported in pregnancies diagnosed with polyhydramnios [2,5,9]; however, there are contrasting results regarding maternal and neonatal outcomes when polyhydramnios is idiopathic [2,5,10,11]. For example, Lallar et al. [12] reported high rates of maternal and perinatal complications of idiopathic polyhydramnios. In fact, they outlined higher rates of preterm delivery (54% in the polyhydramnios group vs. 5.6% in controls), postpartum hemorrhage (6.6% in polyhydramnios vs. 2% in controls), intrauterine deaths (10.4% in polyhydramnios vs. 0.4% in controls), and perinatal deaths (22% in polyhydramnios vs. 0.1% in controls). However, they did not stratify their population by severity. Asadi et al. [5] compared the outcomes between groups with different severities of polyhydramnios, reporting that the incidences of prematurity and low Apgar score were higher in the moderate and severe groups compared to the mild group. They also compared the whole study group with a control population and found that pregnancies with idiopathic polyhydramnios had a higher rate of maternal and feto-neonatal complications. However, they did not compare the outcomes of each polyhydramnios subgroup (mild, moderate, and severe) with controls.

Since the mild idiopathic form represents the majority of cases of polyhydramnios in most studies, our primary aim was to analyze the obstetric and neonatal outcomes in pregnancies diagnosed with mild idiopathic polyhydramnios and to compare them with the outcomes of pregnancies not complicated by this condition to guide management and counseling in clinical practice.

## 2. Materials and Methods

The present retrospective cohort study was conducted in a single university hospital public tertiary care referral center (between 2010 and 2019). The study group of singleton pregnancies (*n* = 109) was compared with the population of healthy women with singleton pregnancies who delivered at the same university hospital (*n* = 2550). The study was approved by the local ethics committee (acceptance number 13738, approved on 27 November 2018). Informed consent was collected from every patient included in the study. Polyhydramnios was defined as the presence of an AFI ≥ 25 cm or a DVP ≥ 8 cm and was stratified into mild, moderate, and severe subgroups. The maximum AFI and/or DVP were considered when multiple ultrasound exams were performed. Idiopathic polyhydramnios was defined as follows: the presence of a normal detailed ultrasound scan and a normal karyotype and the absence of genetic syndromes diagnosed prenatally or postnatally, gestational or pregestational diabetes, and maternal–fetal infections. Maternal and neonatal follow-ups were obtained from medical records and patient interviews. Cases with unknown pregnancy or perinatal outcomes and those who did not give consent to the study were excluded. The maternal age at the time of delivery, parity, pregestational body mass index (BMI), mode of conception, and sex of the newborns were recorded. The following obstetric outcomes were assessed: rate of induction of labor, gestational age at delivery, mode of delivery, Cesarean section (CS) rate and indication, incidence of labor complications, and postpartum hemorrhage (defined as minor if blood loss was between 500 and 1000 cc in spontaneous delivery and as major if blood loss was ≥1000 cc). Fetal malpresentations, labor dystocia, abnormal cardiotocography (CTG), failure to progress, umbilical cord prolapse, and placental abruption were the labor complications that were investigated. Perinatal outcomes included: intrauterine and perinatal mortality, neonatal hypoxia (defined as a pH < 7.10), 1 and 5 min Apgar scores ≤ 7, and birth weight. Macrosomia was defined as a neonatal birth weight ≥ 4500 gr. Finally, when more than one ultrasound scan was performed, we assessed whether mild polyhydramnios returned to normal AFI/DVP values, progressed to a higher degree, or remained stable.

The statistical analysis was performed using SPSS version 21.0 (SPSS Inc., Chicago, IL, USA) and the Microsoft Office Excel package (Microsoft Corp., Redmond, WA, USA). Chi-square or Fisher’s exact tests were used to compare categorical variables and *t*-tests were used for continuous variables. Significance was considered at the level of *p* < 0.05. Odds ratios (ORs) and 95% confidence intervals were also calculated.

## 3. Results

This study included 356 singleton pregnancies complicated by polyhydramnios; among them, 177 (49.7%) cases were idiopathic, of which 109 (61.6%) were defined as mild cases.

### 3.1. Maternal Characteristics

The maternal characteristics of the study population are reported in Table 1. Patients with mild idiopathic polyhydramnios were older than controls (*p* < 0.001). Parity, the pregestational body mass index (BMI), and the mode of conception did not differ between the study group and the controls. There were more male fetuses in the polyhydramnios group than in the control group (68.9% vs. 51%; *p* = 0.001).

### 3.2. Obstetric Outcomes

Table 2 reports the obstetric outcomes of the polyhydramnios and control groups. In the polyhydramnios group, the overall rate of CS was significantly increased (*p* = 0.047), and this increase was attributed to a higher rate of emergency c-section in the polyhydramnios group compared to controls. When the indications to CS were compared, the group with polyhydramnios had higher incidences of abnormal CTG (7.3% vs. 2.9%; *p* = 0.018) and labor dystocia (8.2% vs. 2.9%; *p* = 0.006) compared to controls (Table 3). Overall, the rate of CS performed for labor complications was statistically higher in the polyhydramnios group than in controls (28.44% vs. 17.84%, *p* = 0.007) (Table 3). No differences were found in the rates of labor induction (29.4% vs. 23.9%; *p* = 0.21) and spontaneous preterm delivery (3.67% vs. 3.05%; *p* = 0.58) or the gestational age at delivery (39.17 ± 1.77 weeks vs. 39.2 ± 6.4 weeks; *p* = 0.61) between the two groups (Table 2). Women with polyhydramnios showed differences in terms of postpartum hemorrhage, umbilical cord prolapse, placental abruption, shoulder dystocia, and fetal malpresentations.

The evolution of polyhydramnios was assessed in 75/109 cases who had more than one ultrasound evaluation in our department. The AFI returned to normal values in 51 cases (68%, 51/75), remained stable in 22 cases (29.3%, 22/75), and progressed to moderate polyhydramnios in 2 cases (2.6%, 2/75).

### 3.3. Perinatal Outcomes

Mild idiopathic polyhydramnios was not associated with an increased rate of intrauterine or neonatal death (0% vs. 0.31%; *p* = 1.00). With regards to neonatal characteristics, no differences in the Apgar score at 1 and 5 min or in the rate of neonatal hypoxia were observed between pregnancies complicated by mild polyhydramnios and the control group, as shown in Table 4. The incidence of macrosomia was higher in the mild idiopathic polyhydramnios group than in controls, but the difference was not significant (*p* = 0.10).

In four cases, additional diagnoses were made after birth: one case of bilateral deafness, one case of vesicoureteral reflux, one case of kidney stones, and one case of hyperphenylalaninemia.

## 4. Discussion

Our data show that patients with pregnancies complicated by mild idiopathic polyhydramnios have an increased risk of emergency CS but may be reassured with regards to the other possible obstetric and perinatal complications. 

Generally, pregnancies complicated by polyhydramnios are considered to be at increased risk of adverse obstetric outcomes. However, there is no consensus about the risks associated with mild idiopathic polyhydramnios. Moreover, despite the presence of several studies on idiopathic polyhydramnios in the literature, some of them did not stratify cases by severity and others did not include a control population.

The finding of an increased rate of CS in our study group is in accordance with other authors. Dorleijn et al. [13] found an increased rate of CS in the idiopathic polyhydramnios group compared to the general population. In their study, however, pregnancies were not stratified based on polyhydramnios severity, and the rate of mild polyhydramnios was not reported. These results were also confirmed by Yefet et al. [14], who reported an increased rate of CS, mostly due to elective surgery for suspected macrosomia. Wiegand et al. [15] reported increased rates of Cesarean delivery and postpartum hemorrhage in idiopathic polyhydramnios, and these data were significant, even after adjustment for polyhydramnios severity. In our study, the increased rate of emergency CS was mainly due to abnormal CTG and labor dystocia. Different from previous reports, we did not observe an increased rate of CS for fetal malposition, cephalo-pelvic disproportion, or macrosomia [13,14,15].

The incidences of preterm delivery and postpartum hemorrhage were not increased in our population. Currently, studies on these outcomes in women with polyhydramnios report conflicting results. Wiegand et al. [15] did not find an increased rate of preterm birth within the idiopathic polyhydramnios group; their reported rate of preterm birth in patients with polyhydramnios (11.8%) was consistent with the rate of preterm delivery in the United States. On the other hand, Dorleijn et al. [13] observed a higher incidence of preterm deliveries (20.5%) in cases with polyhydramnios. Again, caution should be used because of the lack of stratification according to polyhydramnios severity.

Considering our results, patients with mild idiopathic polyhydramnios should be reassured about their obstetric risks. Moreover, in our study the progression of mild idiopathic polyhydramnios to a higher degree of severity occurred in only 2.6% of cases, while in 68% of cases the amniotic fluid returned to normal values. These data suggest that mild stable polyhydramnios should be considered a different condition from moderate and severe polyhydramnios. In addition, patients with stable polyhydramnios should be even more reassured since most fetal pathologies are characterized by a worsening of the condition.

With regards to neonatal outcomes, Abele et al. [16] reported that the postnatal examination of 118 fetuses revealed a prenatally unexpected anomaly in 11 cases (9.3%) and that the severity of polyhydramnios or other antenatal characteristics did not help to detect these anomalies before birth. On the contrary, Dashe et al. [17] found that the probability of a major fetal anomaly increased with the amniotic fluid volume, from 1% in the case of mild polyhydramnios to 11% in the case of severe polyhydramnios. This was also confirmed by Belthold et al. [18], who reported that severe polyhydramnios was associated with an unfavorable postnatal outcome compared to the mild group. 

In our study, there were no cases of intrauterine or neonatal death, and the number of cases where additional diagnoses were revealed after birth was low (3.2%). Moreover, these conditions were probably not related to the prenatally diagnosed mild idiopathic polyhydramnios. We could therefore suggest that patients with pregnancies complicated by mild idiopathic polyhydramnios could be reassured regarding feto-neonatal outcomes, which appear to be comparable to the general population.

An interesting observation of our present study is that mild idiopathic polyhydramnios was more frequent in male fetuses compared with females (69% vs. 31%). The exact pathophysiology of this finding remains unknown, and several factors may be involved. For example, it could be associated with a higher weight in males compared to females. Stanescu et al. [19] stated that the level of amniotic fluid was increased in males independent of fetal biometry. This result suggests that mild idiopathic polyhydramnios, especially in male fetuses, has a different pathophysiology than the moderate and severe forms and could be considered a paraphysiological condition.

The strengths of our study are that all ultrasound scans were performed in the same ultrasound unit and that all patients in the polyhydramnios and control groups were managed and delivered in the same hospital with homogeneous management of pregnancy, labor, and delivery. Moreover, the control group was composed of consecutive women who delivered in our hospital. Therefore, they can be considered representative of the general population; this avoids selection bias that could modify the prevalence of pregnancy complications.

In conclusion, mild idiopathic polyhydramnios should be considered a different clinical condition in comparison with the moderate and severe forms since, as reported in literature, these conditions have worse neonatal outcomes. When this complication occurs during pregnancy, women should be reassured regarding maternal and feto-neonatal outcomes, as the only increased risk identified in our study was that of an emergency CS. Therefore, the management of pregnancies where mild idiopathic polyhydramnios is stable should not be different from pregnancies not complicated by this condition, except for the need for increased labor surveillance.

## Figures and Tables

**Table 1 children-09-01624-t001:** Maternal characteristics of the study population.

Demographic Characteristics	Mild Idiopathic Polyhydramnios (*n* = 109)	Controls (*n* = 2550)	*p*-Values
Mean maternal age (±SD)	35.6 ± 5.5	31.8 ± 6.1	<0.001 *
Nulliparity	49 (45%)	1384 (54.3%)	0.7525
Body mass index	22.5 ± 3.8	22.5 ± 1.6	1.000
Assisted reproductive technology	5 (4.6%)	136 (5.3%)	1.000
Male gender	62 (56.9%)	1301 (51%)	0.001 *

* The sign points out that the data is significant.

**Table 2 children-09-01624-t002:** Obstetric outcomes of the study population and control group.

Obstetric Outcomes	Mild Idiopathic Polyhydramnios (*n* = 109)	Controls (*n* = 2550)	*p*-Values	OR (IC 95%)
Cesarean section	50 (45.9%)	825 (32.4%)	0.047 *	1.77 (1.20–2.60)
Elective	22 (20.2%)	374 (14.6%)	0.129	1.47 (0.91–2.37)
Emergency	28 (25.7%)	451 (17.7%)	0.041 *	1.60 (1.03–2.50)
Induction of labor	32 (29.4%)	609 (23.9%)	0.208	1.32 (0.86–2.02)
Spontaneous preterm delivery <37 w	4 (3.7%)	78 (3.1%)	0.577	1.20 (0.43–3.36)
≤34 w	2 (1.8%)	25 (1.0%)	0.304	1.88 (0.44–8.07)
≤32 w	0	21 (0.8%)	1.000	0.53 (0.03–8.92)
Postpartum hemorrhage	10 (9.2%)	337 (13.2%)	0.247	0.66 (0.34–1.28)
Postpartum hemorrhage ≥ 1000 cc	6 (5.5%)	110 (4.3%)	0.474	1.29 (0.55–3.00)
Umbilical cord prolapse	1 (0.9%)	3 (0.1%)	0.159	7.86 (0.81–76.19)
Placental abruption	0	8 (0.3%)	1.000	1.36 0.07–23.81)
Fetal malpresentations	5 (4.6%)	125 (4.9%)	1.000	0.93 (0.37–2.32)

* The sign points out that the data is significant.

**Table 3 children-09-01624-t003:** Indication to Cesarean section in cases and controls.

Indication of Cesarean Delivery	Mild Idiopathic Polyhydramnios (*n* = 109)	Controls (*n* = 2550)	*p*-Values
Macrosomia	1 (0.9%)	11 (0.4%)	0.208
Fetal malposition	6 (5.5%)	81 (3.2%)	0.168
Labor dystocia	9 (8.3%)	74 (2.9%)	0.006 *
Abnormal fetal heart rate	8 (7.3%)	74 (2.9%)	0.018 *
Failure to progress	5 (4.6%)	75 (2.9%)	0.379
Placenta abruption	1 (0.9%)	8 (0.3%)	0.314
Repeated Cesarean section	10 (9.2%)	246 (9.7%)	1.000
Preeclampsia/HEELP	1 (0.9%)	6 (0.2%)	0.254
Intrauterine growth restriction	0	37 (1.5%)	0.402
Other	9 (8.3%)	213 (8.4%)	1.000
Labor complications	31 (28.4%)	455 (17.8%)	0.007 *

HELLP: hemolysis, elevated liver enzymes, and low platelet count. * The sign points out that the data is significant.

**Table 4 children-09-01624-t004:** Neonatal outcomes in the study population and control group.

Perinatal Outcomes	Mild Idiopathic Polyhydramnios (*n* = 109)	Controls (*n* = 2550)	*p*-Values	OR (IC 95%)
Live birth	109 (100%)	2542 (99.7%)	1.000	1.18 (0.06–20.74)
IUD/Neonatal mortality	0%	8 (0.3%)		
pH ≤ 7.1	3 (2.8%)	73 (2.9%)	0.816	0.78 (0.28–2.17)
Apgar score 1° min ≤ 7	7 (6.4%)	160 (6.3%)	0.083	1.58(0.93- 2.68)
Apgar score 5° min ≤ 7	0%	45 (1.8%)	0.767	0.63 (0.15–2.64)
Macrosomia (weight ≥ 4500 gr)	2 (1.8%)	11 (0.4%)	0.096	4.31 (0.94–19.70)

IUD: intrauterine death.

## Data Availability

The data that support the findings of this study are available from the corresponding author upon reasonable request.

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
