# Peer review of "Obstetric and Neonatal Outcomes in Mild Idiopathic Polyhydramnios"

_children, 2022, doi:10.3390/children9111624_

Round 1

Reviewer 1 Report

This reviewer considers this work interesting. I have a few minor comments only.

Line 68: what is n=1378?

Discussion: First and sixth paragraphs are identical. Also, 2nd and 3rd.

Line165: Use Dorlein et al.

Finally, line 199, I think you want to state something else than you did, because you have not studied the difference between mild idiopathic and other, more severe, polyhydramnios. My guess is that being a non-native English speaking group caused this sentence. This reviewer recognizes it because I am neither non-native English speaking.

Author Response

Response to Reviewer 1 Comments

Point 1: Line 68: what is n=1378?.

Response 1: We have clarified that the number is the approval number of ethics committee and we have added the date of approval.

Point 2: Discussion: First and sixth paragraphs are identical. Also, 2nd and 3rd.

Response 2: Thanks for the comment. We have modified paragraph number sixth from the first one. We have deleted 2nd paragraphs since it was doubled and modified 3rd paragraph to make it more complete and accurate.

Point 3: Line165: Use Dorlein et al..

Response 3: We have fixed the typing mistake.

Point 4: Finally, line 199, I think you want to state something else than you did, because you have not studied the difference between mild idiopathic and other, more severe, polyhydramnios. My guess is that being a non-native English speaking group caused this sentence. This reviewer recognizes it because I am neither non-native English speaking.

Response 4: Thank you for the suggestion. We have rewrited the sentence in a clearer form.

Reviewer 2 Report

Dear Authors, I want to congratulate with you for this interesting paper

I think mild polyhydramnios is a very frequent complication of pregnancy that pose many questions and the literature is always not enough therefore well done

The paper is well written and the study well conducted, 

I would just suggest two things

1) To add within the introduction or the discussion the critical role that to rule out GDM has to prevent polyhydramnios, 

A universal screening for GDM will let us capture a significant proportion of GDM women otherwise unknown that may present with this complication

and if GDM will be recognized to properly counsel women regarding GDM related risks (such as polyhydramnios, Macrosomia etc etc) will let them strictly follow recommendation and reduce pregnancy complication related to polyhydramnios

with this aim I suggest to read and cite the following paper  PMID: 34627198

2) please add a table with the summary of findings that may help the reader to immediately capture the meaning of the paper

best regards

Author Response

Response to Reviewer 2 Comments

Point 1: To add within the introduction or the discussion the critical role that to rule out GDM has to prevent polyhydramnios, 

A universal screening for GDM will let us capture a significant proportion of GDM women otherwise unknown that may present with this complication

and if GDM will be recognized to properly counsel women regarding GDM related risks (such as polyhydramnios, Macrosomia etc etc) will let them strictly follow recommendation and reduce pregnancy complication related to polyhydramnios

with this aim I suggest to read and cite the following paper  PMID: 34627198

Response 1: Thanks for this useful suggestion. We have incorporated in the introductin this aspect focusing on the need of universal screening and proper counseling about the condition.

Point 2: please add a table with the summary of findings that may help the reader to immediately capture the meaning of the paper

Response 2: Thanks for the suggestion but we think that the findings are already reported in table 2, 3 and 4. The addition of another table, which would be quite long since there are many analyzed outcomes, could decrease the readability of the paper.